# GRADIENT PROPERTIES OF HARD THRESHOLDING OPERATOR

## ABSTRACT

Sparse optimization receives increasing attention in many applications such as compressed sensing, variable selection in regression problems, and recently neural network compression in machine learning. For example, the problem of compressing a neural network is a bi-level, stochastic, and nonconvex problem that can be cast into a sparse optimization problem. Hence, developing efficient methods for sparse optimization plays a critical role in applications. The goal of this paper is to develop analytical techniques for general, large size sparse optimization problems using the hard thresholding operator. To this end, we study the iterative hard thresholding (IHT) algorithm, which has been extensively studied in the literature because it is scalable, fast, and easily implementable. In spite of extensive research on the IHT scheme, we develop several new techniques that not only recover many known results but also lead to new results. Specifically, we first establish a new and critical gradient descent property of the hard thresholding (HT) operator. Our gradient descent result can be related to the distance between points that are sparse. However, the distance between sparse points cannot provide any information about the gradient in the sparse setting. To the best of our knowledge, the other way around (the gradient to the distance) has not been shown so far in the literature. Also, our gradient descent property allows one to study the IHT when the stepsize is less than or equal to 1/L, where L>0 is the Lipschitz constant of the gradient of an objective function. Note that the existing techniques in the literature can only handle the case when the stepsize is strictly less than 1/L. By exploiting this we introduce and study HT-stable and HT-unstable stationary points and show no matter how close an initialization is to a HT-unstable stationary point (saddle point in sparse sense), the IHT sequence leaves it. Finally, we show that no matter what sparse initial point is selected, the IHT sequence converges if the function values at HT-stable stationary points are distinct, where the last condition is a new assumption that has not been found in the literature. We provide a video of 4000 independent runs where the IHT algorithm is initialized very close to a HT-unstable stationary point and show the sequences escape them.

## 1 INTRODUCTION

Solving sparse problems has gained increasing attention in the fields of statistics, finance, and engineering. These problems emerge in statistics as variable selection in linear regression problems Fan & Li (2001); Zou & Hastie (2005); Chun & Keleş (2010); Desboulets (2018), mixed-integer programs Bourguignon et al. (2015); Liu et al. (2017); Dedieu et al. (2021), portfolio optimization in finance Brodie et al. (2009); Chang et al. (2000), compressed sensing in signal processing Foucart & Rauhut (2013); Eldar & Kutyniok (2012), and compressing deep neural networks in machine learning Damadi et al. (2022); Molchanov et al. (2017); Gale et al. (2019), just to name a few. Due to the use of $\ell_0$-(pseudo) norm[1], these problems are discontinuous and nonconvex. The $\ell_0$-norm case have been addressed by the hard thresholding (HT) techniques specially the iterative HT (IHT) scheme Blumensath & Davies (2008); Beck & Eldar (2013); Lu (2014); Zhou et al. (2021). The Lasso-type, Basic Pursuit(BP)-type, and BP denoising(BPDN)-type problems consider $\ell_1$-norm as a convex approximation of $\ell_0$-norm Tibshirani (1996); Mousavi & Shen (2019). Nonconvex approximation of $\ell_0$-norm as $\ell_p$-(pseudo) norm $(0 < p < 1)$ has also been studied well Chartrand

---

[1] $\ell_0$ is not mathematically a norm because for any norm $\| \cdot \|$ and $\alpha \in \mathbb{R}$, $\|\alpha\boldsymbol{\theta}\| = |\alpha|\|\boldsymbol{\theta}\|$, while $\|\alpha\boldsymbol{\theta}\|_0 = |\alpha|\|\boldsymbol{\theta}\|_0$ if and only if $|\alpha| = 1$

---

**Algorithm 1** The iterative hard thresholding (IHT)

---

**Require:** $\mathbf{x}^0 \in \mathbb{R}^n$ such that $\|\mathbf{x}^0\|_0 \leq s$ and stepsize $\gamma > 0$ .
  1: $\mathbf{x}^{k+1} \in H_s(\mathbf{x}^k - \gamma \nabla f(\mathbf{x}^k))$ for $k = 0, 1, \ldots$

---

(2007); Foucart & Lai (2009); Lai & Wang (2011); Wang et al. (2011); Zheng et al. (2017); Won et al. (2022). Sparse optimization problems can also be formulated as mixed-integer programs Burdakov et al. (2016). Intrinsic combinatorics involved in sparse optimization problems makes it an NP-hard problem (even for a quadratic loss Davis (1994); Natarajan (1995)) so it is difficult to find a global minimizer. However, greedy algorithms have developed to find local minimizers. To this end, following the ideas of matching pursuit (MP) and orthogonal MP (OMP) Mallat & Zhang (1993); Pati et al. (1993) as greedy algorithms, numerous other greedy algorithms have been developed such as stagewise OMP (StOMP) Donoho et al. (2012), regularized OMP (ROMP) Needell & Vershynin (2009; 2010), Compressive Sampling MP (CoSaMP) Needell & Tropp (2009), and Gradient Support Pursuit (GraSP) Bahmani et al. (2013). It should be noted that sparse optimization is not restricted to finding a sparse vector. For example Fornasier et al. (2011); Haeffele et al. (2014); Davenport & Romberg (2016), finding a low-rank matrix is considered. The problem of finding a low-rank matrix is a counterpart to finding a sparse vector when it comes to applications dealing with matrices. In addition to devising algorithms for solving sparse optimization problems, developing first and second order optimality conditions have also been addressed well Pan et al. (2017a); Bauschke et al. (2014); Beck & Hallak (2016); Li & Song (2015); Lu (2015); Pan et al. (2015); Bucher & Schwartz (2018).

The general sparse optimization problem is the following:

$$\begin{aligned} &\min f(\mathbf{x}) \\ &\text{s.t. } C_s \cap \mathcal{X} \end{aligned} \qquad (1)$$

where $C_s = \{\mathbf{x} \in \mathbb{R}^n \mid \|\mathbf{x}\|_0 \leq s\}$ (sparsity constraint) is the union of finitely many subspaces of dimension $s$ such that $1 \leq s < n$, $\mathcal{X}$ is a constraint set in $\mathbb{R}^n$, and the objective function $f : \mathbb{R}^n \to \mathbb{R}$ is lower bounded and continuously differentiable, i.e., $C^1$. In this paper we address a special case of Problem (1) where $\mathcal{X} = \mathbb{R}^n$ as follows:

$$(\mathbf{P}) : \quad \begin{aligned} &\min f(\mathbf{x}) \\ &\text{s.t. } \mathbf{x} \in C_s \end{aligned} \qquad (2)$$

To address Problem (2) the following fundamental questions arise:

(Q1) What are the necessary/sufficient conditions for a local/global minimizer of Problem (2)?

(Q2) What are the characteristics of accumulation points of algorithms solving Problem (2)?

(Q3) Under what condition(s) does an accumulation point become a local/global minimizer?

(Q4) If an accumulation point is a local/global minimizer, what is the rate of convergence?

By considering the IHT algorithm, we will answer the above questions. This algorithm has been extensively studied in the literature. It was originally devised for solving compressed sensing problems in 2008 Blumensath & Davies (2008; 2009). Since then, there has been a large body of literature studying the IHT-type algorithms from different standpoints. For example, Beck & Eldar (2013); Lu (2014; 2015); Pan et al. (2017b); Zhou et al. (2021) consider convergence of iterations, Jain et al. (2014); Liu & Foygel Barber (2020) study the limit of the objective function value sequence, Liu et al. (2017); Zhu et al. (2018) address duality, Zhou et al. (2020); Zhao et al. (2021) extend it to Newton's-type IHT, Chen & Gu (2016); Li et al. (2016); Liang et al. (2020); Zhou et al. (2018) consider the stochastic version, Blumensath (2012); Khanna & Kyrillidis (2018); Vu & Raich (2019); Wu & Bian (2020) address accelerated IHT, and Wang et al. (2019); Bahmani et al. (2013) solve logistic regression problem using the IHT.

SUMMARY OF CONTRIBUTIONS

By considering the IHT Algorithm 1 for Problem (2), we develop the following results:

- We establish a new critical gradient descent property of the hard thresholding (HT) operator that has not been found in the literature. Our gradient descent result can be related to

the distance between points that are sparse. However, the distance between sparse points cannot provide any information about the gradient in the sparse setting. To the best of our knowledge, the other way around (the gradient to the distance) has not been shown so far in the literature. This property allows one to study the IHT when the stepsize is less than or equal to $1/L$, where $L > 0$ is the Lipschitz constant of the gradient of an objective function. Note that the existing techniques in the literature can only handle the case when the stepsize is strictly less than $1/L$. As an example, one can refer to Liu & Foygel Barber (2020) that needs the stepsize to be greater than or equal to $1/L$.

- We introduce the notion of *HT-stable/unstable stationary* points. Using them we establish the *escapability* property of *HT-unstable stationary* points (saddle point the sparse sense) and local *reachability* property of strictly *HT-stable stationary* points. We provide a video of 4000 independent runs where the IHT algorithm is initialized very close to a HT-unstable stationary point and show the sequences escape them.

- We also show that the IHT sequence converges globally under a new assumption that has not been found in the literature. In addition, Q-linearly convergence of the IHT algorithm towards a local minimum when the objective function is both RSS and restricted strictly convex is shown.

According to our results, we address (Q1) and (Q2) by establishing a new gradient descent property of the hard thresholding (HT) operator and introducing the notion of *HT-stable/unstable stationary* points. By considering RSS, restricted strictly convex, and RSC properties we address (Q3) and (Q4). Table 1 is provided to compare our results with those in the literature. It shows what has been done chronologically and demonstrates our results.

## 2 RELATED WORK

To answer (Q1) Beck & Eldar (2013) introduces *L-stationarity* property as a necessary condition for an optimal solution of Problem (2). The *L-stationarity* property is defined when the gradient of the objective function is Lipschitz. Also, Beck & Eldar (2013) addresses (Q2) by showing any accumulation point of the IHT algorithm is *L*-stationary. Lu in Lu (2014) restricts the objective function to be convex and shows that the IHT sequence converges to a local minimum when the objective function is regularized by $\ell_0$-norm and $\mathcal{X}$ is a box constraint. Jain et al. Jain et al. (2014) put more restriction on the objective value function and show that the objective value function sequence generated by the IHT algorithm converges to a value attained under a more restricted sparsity constraint. The restrictions used in Jain et al. (2014) are Restricted Strong Smoothness (RSS) and Restricted Strong Convexity (RSC). The RSS and RSC properties are introduced by Negahban et al. (2012) and first used by Bahmani et al. (2013) for sparsity optimization problems. Currently, they have become standard restrictions for analyzing sparsity optimization problems. Under RSS and RSC properties for the objective function, one is able to address (Q3) and (Q4).

Finding a closed-form expression for $P_{C_s \cap \mathcal{X}}$ when $\mathcal{X}$ is an arbitrary set is difficult. However, Beck & Hallak (2016) shows orthogonal projection of a point onto $C_s \cap \mathcal{X}$ can be efficiently computed when $\mathcal{X}$ is a symmetric closed convex set. In this context, two types of sets are of interest: nonnegative symmetric sets and sign free sets. To address (Q1) in a more generalized setting, Beck and Hallak Beck & Hallak (2016) characterize *L-stationary* points of Problem (1) when $\mathcal{X}$ is either nonnegative symmetric set or sign free. Also, Lu in Lu (2015) considers the same setting as Beck & Hallak (2016) and introduces a new optimality condition that is stronger than *L-stationary*. He devises a Nonmonotone Projected Gradient (NPG) algorithm and shows an accumulation of the NPG sequence is the global optimal of Problem (1). Pan et al. Pan et al. (2017a) consider Problem (1) when $\mathcal{X} = \mathbb{R}^n_+$. They develop an Improved IHT algorithm (IIHT) that employs the Armijo-type stepsize rule. They show when the objective function is RSS and RSC, the IIHT sequence converges to a local minimum. A recent work by Zhou et al. Zhou et al. (2021) develops Newton Hard-Thresholding Pursuit (NHTP) for solving problem (2). They show that when accumulation points of the NHTP sequence are *L-stationary* and are isolated, the sequence converges with a locally Q-quadratic rate. Table 1 compares current results in the literature.

**Table 1:** Comparison of results for the deterministic IHT-type algorithms.

| Paper | Objective function | Constraints | Stepsize | Optimality conditions | Method | Convergence |
|---|---|---|---|---|---|---|
| Beck & Eldar (2013) | $f(\mathbf{x})$: $L$-LG | $\mathbf{x} \in C_s$ | $0 < \gamma < 1/L$ | $\mathbf{x}^* \in H_s(\mathbf{x}^* - \frac{1}{L}\nabla f(\mathbf{x}^*))$ | IHT | Any accumulation point of the IHT sequence satisfies $\mathbf{x}^* \in H_s(\mathbf{x}^* - \frac{1}{L}\nabla f(\mathbf{x}^*))$(Theorem 3.1). |
| Lu (2014) | $f(\mathbf{x}) + \lambda\|\mathbf{x}\|_0$ $f(\mathbf{x})$ is convex | $\mathbf{x} \in C_s$ $l \leq \mathbf{x} \leq u$ | $0 < \gamma < 1/L$ | Local minimizer | | The IHT sequence converges to a local minimizer. (Theorem 3.3) |
| Beck & Hallak (2016) | $f(\mathbf{x})$: $L$-LG | $\mathbf{x} \in C_s \cap \mathcal{X}$, $\mathcal{X}$ is nonnegative symmetric or sign free | $0 < \gamma < 1/L$ | Basic feasibility | BFS | The sequence of BFS converges to a basic feasible point (Lemma 7.1). |
| Lu (2015) | $f(\mathbf{x})$: $L$-LG | $\mathbf{x} \in C_s \cap \mathcal{X}$, $\mathcal{X}$ is nonnegative symmetric or sign free | $0 < \gamma < 1/L$ | $\mathbf{x}^* = P_{C_s \cap \mathcal{X}}(\mathbf{x}^* - \frac{1}{L}\nabla f(\mathbf{x}^*))$ | NPG | An accumulation point of NPG sequence satisfies the optimality condition (Theorem 4.3). |
| Pan et al. (2017b) | $f(\mathbf{x})$ $L_s$-RSS & $s$-RC | $\mathbf{x} \in C_s \cap \mathbb{R}^n_+$ | $0 < \gamma < 1/L_s$ | $\mathbf{x}^* \in P_{C_s \cap \mathbb{R}^n_+}(\mathbf{x}^* - \frac{1}{L}\nabla f(\mathbf{x}^*))$ or $L_s$-stationary | IIHT | Any accumulation point of IIHT sequence converges to a $L_s$-stationary point (Theorem 3.1). |
| Pan et al. (2017b) | $f(\mathbf{x})$ $L_s$-RSS & $\beta_s$-RSC | $\mathbf{x} \in C_s \cap \mathbb{R}^n_+$ | $0 < \gamma < 1/L_s$ | $\mathbf{x}^* \in P_{C_s \cap \mathbb{R}^n_+}(\mathbf{x}^* - \frac{1}{L}\nabla f(\mathbf{x}^*))$ or $L_s$-stationary | IIHT | The IIHT sequence converges to a local minimizer (Theorem 3.2). If $\|\mathbf{x}^*\|_0 < s$, then $\mathbf{x}^*$ is a global minimizer. When $\|\mathbf{x}^*\|_0 = s$, the IIHT sequence converges Q-linearly (Theorem 3.4). |
| Zhou et al. (2021) | $f(\mathbf{x})$ $L_s$-RSS | $\mathbf{x} \in C_s$ | $0 < \gamma < 1/L_s$ | $\mathbf{x}^* \in H_s(\mathbf{x}^* - \frac{1}{L_s}\nabla f(\mathbf{x}^*))$ or $L_s$-stationary | NHTP | Any accumulation point $\mathbf{x}^*$ of the NHTP sequence is an $L_s$-stationary point (Theorem 9). If $\mathbf{x}^*$ is isolated, the entire sequence converges. |
| Zhou et al. (2021) | $f(\mathbf{x})$ $L_s$-RSS & $\beta_s$-RSC & Restricted Hessian is Lipschitz | $\mathbf{x} \in C_s$ | $0 < \gamma < 1/L_s$ | $\mathbf{x}^* \in H_s(\mathbf{x}^* - \frac{1}{L_s}\nabla f(\mathbf{x}^*))$ or $L_s$-stationary | NHTP | The NHTP sequence converges to a $L_s$-stationary point (Theorem 10). Locally, it converges quadratically. |
| Ours | $f(\mathbf{x})$ $L_s$-RSS | $\mathbf{x} \in C_s$ | $0 < \gamma \leq 1/L_s$ | $\mathbf{x}^* \in H_s(\mathbf{x}^* - \frac{1}{L_s}\nabla f(\mathbf{x}^*))$ or HT-stable | IHT | If $f(\mathbf{x}^*) \neq f(\mathbf{y}^*)$ for all $\mathbf{x}^*, \mathbf{y}^*$ HT-stable points, by starting from a $\mathbf{x}^0 \in C_s$ the IHT sequence converges to some HT-stable points. Corollary 5: global convergence |
| Ours | $f(\mathbf{x})$ $L_s$-RSS | $\mathbf{x} \in C_s$ | $0 < \gamma < 1/L_s$ | $\mathbf{x}^* \in H_s(\mathbf{x}^* - \frac{1}{L_s}\nabla f(\mathbf{x}^*))$ or HT-stable | IHT | If all HT-stable points are isolated, by starting from a $\mathbf{x}^0 \in C_s$ the IHT sequence converges to some HT-stable points. Corollary 5: global convergence |
| Ours | $f(\mathbf{x})$ $L_s$-RSS Strictly convex on $\mathcal{S}_{\mathcal{J}}$ with $|\mathcal{J}| = s$ | $\mathbf{x} \in C_s$ | $0 < \gamma \leq 1/L_s$ | $\mathbf{x}^* = H_s(\mathbf{x}^* - \frac{1}{L_s}\nabla f(\mathbf{x}^*))$ or HT-strictly stable | IHT | By starting from $\mathbf{x}^0 \in \mathcal{B}(\mathbf{x}^*, \delta) \cap C_s$ the IHT sequence converges to HT-strictly stable $\mathbf{x}^*$ point Q-linearly (Proposition 1). |

## 3 DEFINITIONS

We provide some definitions that will be used throughout the paper. These definitions are the HT operator (HTO) and HTO inequality, RSS and RSC functions.

**Definition 1** (Restricted Strong Smoothness (RSS)). *A differentiable function $f : \mathbb{R}^n \to \mathbb{R}$ is said to be restricted strongly smooth with modulus $L_s > 0$ or is $L_s$-RSS if*

$$f(\mathbf{y}) \leq f(\mathbf{x}) + \langle \nabla f(\mathbf{x}), \mathbf{y} - \mathbf{x} \rangle + \frac{L_s}{2}\|\mathbf{y} - \mathbf{x}\|_2^2 \quad \forall \mathbf{x}, \mathbf{y} \in \mathbb{R}^n \text{ such that } \|\mathbf{x}\|_0 \leq s, \|\mathbf{y}\|_0 \leq s. \tag{3}$$

**Definition 2** (Restricted Strong Convexity (RSC)). *A differentiable function $f : \mathbb{R}^n \to \mathbb{R}$ is said to be restricted strongly convex with modulus $\beta_s > 0$ or is $\beta_s$-RSC if*

$$f(\mathbf{y}) \geq f(\mathbf{x}) + \langle \nabla f(\mathbf{x}), \mathbf{y} - \mathbf{x} \rangle + \frac{\beta_s}{2}\|\mathbf{y} - \mathbf{x}\|_2^2 \quad \forall \mathbf{x}, \mathbf{y} \in \mathbb{R}^n \text{ such that } \|\mathbf{x}\|_0 \leq s, \|\mathbf{y}\|_0 \leq s. \tag{4}$$

**Definition 3** (The HT operator). *The HT operator $H_s(\cdot)$ denotes the orthogonal projection onto multiple subspaces of $\mathbb{R}^n$ with dimension $1 \leq s < n$, that is,*

$$H_s(\mathbf{x}) \in \arg\min_{\|\mathbf{z}\|_0 \leq s} \|\mathbf{z} - \mathbf{x}\|_2. \tag{5}$$

**Claim 1.** *The HT operator keeps the $s$ largest entries of its input in absolute values.*

For a vector $\mathbf{x} \in \mathbb{R}^n$, $\mathcal{I}_s^{\mathbf{x}} \subset \{1, \dots, n\}$ denotes the set of indices corresponding to the first $s$ largest elements of $\mathbf{x}$ in absolute values. For example $H_2([1, -3, 1]^\top)$ is either $[0, -3, 1]^\top$ or $[1, -3, 0]^\top$ where $\mathcal{I}_2^{\mathbf{y}} = \{2, 3\}$ and $\mathcal{I}_2^{\mathbf{y}} = \{1, 2\}$, respectively. Therefore, the output of it may not be unique. This clearly shows why HTO is not a convex operator and why there is an inclusion in (5) not an inequality.

## 4 RESULTS

We consider solving Problem (2) using the IHT Algorithm 1 and develop results on the HT operator. Using them, the behavior of the IHT sequence generated by Algorithm 1 is characterized. Towards

this end, statements of the main results are provided and all the technical proofs are postponed to the Appendix for the reviewers.

## 4.1 GRADIENT DESCENT PROPERTY

First, we establish a new and critical gradient descent property of the hard thresholding (HT) operator.

**Theorem 1.** *Let $f : \mathbb{R}^n \to \mathbb{R}$ be a differentiable function that is $L_s$-RSS, $\mathbf{y} \in H_s(\mathbf{x} - \gamma \nabla f(\mathbf{x}))$ with any $\mathcal{I}_s^{\mathbf{y}}$ and $0 < \gamma \leq \frac{1}{L_s}$, and $\mathbf{x}$ be a sparse vector such that $\|\mathbf{x}\|_0 \leq s$ with any $\mathcal{I}_s^{\mathbf{x}}$. Then,*

$$\frac{\gamma}{2}(1 - L_s\gamma)\|\nabla_{\mathcal{I}_s^{\mathbf{x}} \cup \mathcal{I}_s^{\mathbf{y}}} f(\mathbf{x})\|_2^2 \leq f(\mathbf{x}) - f(\mathbf{y}) \tag{6}$$

*where $\nabla_{\mathcal{I}_s^{\mathbf{x}} \cup \mathcal{I}_s^{\mathbf{y}}} f(\mathbf{x})$ is the restriction of the gradient vector to the union of the index sets $\mathcal{I}_s^{\mathbf{x}}$ and $\mathcal{I}_s^{\mathbf{y}}$.*

Theorem 1 provides a lower bound on the difference between the current function value evaluated at $\mathbf{x}$ and the one evaluated at the updated point provided by the HTO, i.e., $\mathbf{y}$. Note that, $\mathbf{y}$ may not be a unique vector that has $s$ nonzero elements. Nonetheless, as stated in Theorem 1, Inequality (1) holds for any $\mathbf{y}$ that might be the output of the HTO. As one clearly see, the descent can only be characterized by looking at the entries of the gradient that are restricted to the union of the $s$ largest elements in both $\mathbf{x}$ and $\mathbf{y}$. The rest of the gradient can be ignored. Since one may be interested in characterizing the descent using the distance between $\mathbf{x}$ and $\mathbf{y}$, we provide the following corollary.

**Corollary 1.** *Assume all the assumptions in Theorem 1 hold, then,*

$$\frac{1 - L_s\gamma}{6\gamma}\|\mathbf{y} - \mathbf{x}\|_2^2 \leq \frac{\gamma}{2}(1 - L_s\gamma)\|\nabla_{\mathcal{I}_s^{\mathbf{x}} \cup \mathcal{I}_s^{\mathbf{y}}} f(\mathbf{x})\|_2^2 \leq f(\mathbf{x}) - f(\mathbf{y}) \tag{7}$$

The above result shows the superiority of our gradient result because our gradient result can be related to the distance of points that are sparse. However, the distance between sparse points cannot provide any information about the gradient. To the best of our knowledge, the other way around (the gradient to the distance) has not been shown so far in the literature.

Algebraically speaking, characterizing a descent of the function value solely with the information of the current iterate, i.e., $\mathbf{x}$, is of more interest. To this end, we provide another corollary to Theorem 1 that ties the descent to $\mathbf{x}$ only.

**Corollary 2.** *Assume all the assumptions in Theorem 1 hold. Then, the norm of the gradient restricted to any $\mathcal{I}_s^{\mathbf{x}}$ can be bounded as follows:*

$$\frac{\gamma}{2}\|\nabla_{\mathcal{I}_s^{\mathbf{x}}} f(\mathbf{x})\|_2^2 \leq f(\mathbf{x}) - f(\mathbf{y}) \tag{8}$$

By this point, we have shown that applying the HTO once, can result in smaller function value provided the gradient over the $s$ largest entries of $\mathbf{x}$ are nonzero. This can be utilized to show the sequence generated by the IHT algorithm is nonincreasing. Specially, if the generated sequence has an accumulation point, the objective value function sequence converges to the objective value of the accumulation point. [2]

**Corollary 3.** *Let $f : \mathbb{R}^n \to \mathbb{R}$ be a bounded below differential function that is $L_s$-RSS and $(\mathbf{x}^k)_{k \geq 0}$ be the IHT sequence $(\mathbf{x}^k)_{k \geq 0}$ with $0 < \gamma \leq \frac{1}{L_s}$. Then, $(f(\mathbf{x}^k))_{k \geq 0}$ is nonincreasing and converges. Also, if $\mathbf{x}^*$ is an accumulation point of $(\mathbf{x}^k)_{k \geq 0}$ then $(f(\mathbf{x}^k))_{k \geq 0} \to f(\mathbf{x}^*)$.*

Next, we look at *basic stationary points* of Problem (2) and show their properties.

## 4.2 OPTIMALITY CONDITION BASED ON THE HT PROPERTIES

In this subsection, we will show that not all *basic stationary* points of Problem (2) are reachable when the IHT algorithm is run. To do so, the notion of *HT stationary* points are introduced as follows.

---

[2]A sequence may not converge but it may have an accumulation point. For example $1, -1, 1, -1, \ldots$ is not a convergent sequence but it has two accumulation points.

### 4.2.1 HT STATIONARY POINTS

**Definition 4.** *For a given constant $\gamma > 0$, we say that a sparse vector $\mathbf{x}^* \in C_s$ is HT-stable stationary point of Problem (2) associated with $\gamma$ if $\nabla_{supp(\mathbf{x}^*)} f(\mathbf{x}^*) = 0$, and*

$$\min\left(|x_i^*| : i \in \mathcal{I}_s^{\mathbf{x}^*}\right) \geq \gamma \max\left(|\nabla_j f(\mathbf{x}^*)| : j \notin supp(\mathbf{x}^*)\right) = \gamma \|\nabla_{(supp(\mathbf{x}^*))^c} f(\mathbf{x}^*)\|_\infty. \quad (9)$$

*(Note that $\min\left(|x_i^*| : i \in \mathcal{I}_s^{\mathbf{x}^*}\right)$ is unique and does not depend on the choice $\mathcal{I}_s^{\mathbf{x}^*}$.) If $\nabla_{supp(\mathbf{x}^*)} f(\mathbf{x}^*) = 0$ but (9) fails, we say that $\mathbf{x}^*$ is HT-unstable stationary point with $\gamma$. Moreover, if $\nabla_{supp(\mathbf{x}^*)} f(\mathbf{x}^*) = 0$ and (9) holds strictly, namely,*

$$\min\left(|x_i^*| : i \in \mathcal{I}_s^{\mathbf{x}^*}\right) > \gamma \max\left(|\nabla_j f(\mathbf{x}^*)| : j \notin supp(\mathbf{x}^*)\right) \quad (10)$$

*we say that $\mathbf{x}^*$ is a strictly HT-stable stationary point associated with $\gamma$.*

Note that when $\|\mathbf{x}^*\|_0 = s$, $\mathcal{I}_s^{\mathbf{x}^*}$ is unique and equals $supp(\mathbf{x}^*)$ such that $supp(\mathbf{x}^*)$ in the above definition can be replaced by $\mathcal{I}_s^{\mathbf{x}^*}$. Moreover, if $\mathbf{x}^*$ is a strictly *HT-stable* stationary point, then we must have $\mathcal{I}_s^{\mathbf{x}^*} = supp(\mathbf{x}^*)$ (or equivalently $\|\mathbf{x}^*\|_0 = s$) because otherwise, $0 = \min\left(|x_i^*| : i \in \mathcal{I}_s^{\mathbf{x}^*}\right) > \gamma \|\nabla_{(supp(\mathbf{x}^*))^c} f(\mathbf{x}^*)\|_\infty$ which is impossible.

**Remark 1.** *As stated in the Definition 4, a basic stationary point is a point whose gradient is zero over the nonzero elements. For example, suppose $\tilde{\mathbf{x}} = [0, 4, 0, 2]^\top \in \mathbb{R}^4$ is a basic stationary point. Then $\nabla f(\tilde{\mathbf{x}}) = [c_1, 0, c_2, 0]^\top$ where $c_1, c_2$ are scalars. The main idea of the HT-stable stationary point is that it has to be a basic stationary point. In other words $\tilde{\mathbf{x}}$ can be a basic stationary point but not a HT-stable stationary point. This is the analogue of the non-sparse optimization where a point $\hat{\mathbf{x}}$ whose gradient is zero, i.e., $\nabla f(\hat{\mathbf{x}}) = 0$ may not be necessary a local or global minimizer. It can be a saddle point.*

**Remark 2.** *The main message of Definition 4 is the following: "only by looking at the gradient restricted to the nonzero entries of a basic feasible point, one cannot say whether it is a local minimizer of Problem (2) or not".*

An *HT-stable* stationary point associated with $\gamma$ is equivalent to the $\frac{1}{\gamma}$-stationary point of Problem (2) defined in (Beck & Eldar, 2013, Definition 2.3). Thus, by (Beck & Eldar, 2013, Lemma 2.2), $\mathbf{x}^*$ is a *HT-stable* point if and only if $\mathbf{x}^* \in H_s(\mathbf{x}^* - \gamma \nabla f(\mathbf{x}^*))$. The notion of a HT-unstable stationary point is novel and is a key point for proving Theorem 2. Theorem 2 is the foundation for the proof of part b) of Theorem 3 as well as Theorem 4 which characterizes the accumulation point of the IHT sequence. In addition, we have introduced another stationary point, namely strictly HT-stable stationary which is a crucial concept for local convergence of the IHT sequence.

In the following, we present a result that characterizes a *HT-unstable* stationary point. In essence, the following result shows that there always exists a neighborhood around a sparse *HT-unstable* stationary point whose gradient is zero over the nonzero elements and one can decrease the function value by going towards the direction of any nonzero coordinates.

**Theorem 2.** *Suppose $f : \mathbb{R}^n \to \mathbb{R}$ is $C^1$ and $L_s$-RSS. Given any $0 < \gamma \leq \frac{1}{L_s}$, if a vector $\tilde{\mathbf{x}} \in C_s$ is such that $\nabla_{supp(\tilde{\mathbf{x}})} f(\tilde{\mathbf{x}}) = 0$ and $\min\left(|\tilde{x}_i| : i \in \mathcal{I}_s^{\tilde{\mathbf{x}}}\right) < \gamma \|\nabla_{(supp(\tilde{\mathbf{x}}))^c} f(\tilde{\mathbf{x}})\|_\infty$ for some $\mathcal{I}_s^{\tilde{\mathbf{x}}}$, then there exist a constant $\nu > 0$ and a neighborhood $\mathcal{N}$ of $\tilde{\mathbf{x}}$ such that $f(\mathbf{y}) \leq f(\mathbf{x}) - \nu$ for any $\mathbf{x} \in \mathcal{N} \cap C_s$ and any $\mathbf{y} \in H_s(\mathbf{x} - \gamma \nabla f(\mathbf{x}))$.*

The above result leads to the following necessary optimality conditions for a global minimizer of Problem (2) in terms of hard thresholding operator $H_s$. For the case where $\gamma = 1/L_s$, i.e., part b), one needs to use Theorem 2. Indeed, to the best of our knowledge, no proof has not been found for it in the literature. Essentially, establishing the result in part b) is one of our contributions. For the case $\gamma < 1/L_s$ it is proven that $\mathbf{x}^* = H_s(\mathbf{x}^* - \gamma \nabla f(\mathbf{x}^*))$. Note that the condition for $0 < \gamma < \frac{1}{L_s}$ has been obtained in (Beck & Eldar, 2013, Theorem 2.2) without using gradient properties of the HT operator.

**Theorem 3.** *Suppose $f : \mathbb{R}^n \to \mathbb{R}$ is $L_s$-RSS and $\mathbf{x}^*$ is a global minimizer. Then, $\mathbf{x}^*$ is a HT-stable (or $\frac{1}{\gamma}$-) stationary point for any $0 < \gamma \leq \frac{1}{L_s}$. Particularly, the following hold:*

    *a ) For any $0 < \gamma < \frac{1}{L_s}$, $\mathbf{x}^* = H_s(\mathbf{x}^* - \gamma \nabla f(\mathbf{x}^*))$.*

    *b ) For $\gamma = \frac{1}{L_s}$, $\mathbf{x}^* \in H_s(\mathbf{x}^* - \gamma \nabla f(\mathbf{x}^*))$.*

The following result shows that any accumulation point of an IHT sequence must be a *HT-stable* stationary point.

**Theorem 4.** *Let $f : \mathbb{R}^n \to \mathbb{R}$ be $L_s$-RSS and $C^1$ function. Suppose $f$ is bounded below on $C_s$. Consider an IHT sequence $(\mathbf{x}^k)_{k \geq 0}$ associated with an arbitrary $\gamma \in (0, \frac{1}{L_s}]$, and let $\mathbf{x}^*$ be an accumulation point of $(\mathbf{x}^k)_{k \geq 0}$. Then, $\mathbf{x}^*$ is a HT-stable stationary point of Problem (2).*

**Remark 3.** The above theorem shows that any accumulation point of an IHT sequence is a *HT-stable* stationary point of Problem (2). Since each *HT-stable* stationary point must be a basic stationary point, one can observe that any accumulating point $\mathbf{x}^*$ of an IHT sequence must satisfy $\nabla_{\text{supp}(\mathbf{x}^*)} f(\mathbf{x}^*) = 0$ when $\|\mathbf{x}^*\|_0 = s$, or $\nabla f(\mathbf{x}^*) = 0$ when $\|\mathbf{x}^*\|_0 < s$.

The following result pertains to the objective function values of *HT-stable* and *HT-unstable stationary* points.

**Corollary 4.** *Let $f : \mathbb{R}^n \to \mathbb{R}$ be $L_s$-RSS and $C^1$ function. Suppose that every (nonempty) sub-level set of $f$ contained in $C_s$ is bounded, i.e., for any $\alpha \in \mathbb{R}$, $\{x \in C_s | f(\mathbf{x}) \leq \alpha\}$ is bounded (and closed). Consider an arbitrary $\gamma \in (0, \frac{1}{L_s}]$. For any HT-unstable stationary point $\mathbf{x}^*$ associated with $\gamma$, there exists a HT-stable stationary point $\tilde{\mathbf{x}}^*$ associated with $\gamma$ such that $f(\mathbf{x}^*) > f(\tilde{\mathbf{x}}^*)$.*

Based on the above corollary, it is easy to see that if there are finitely many *HT-unstable* stationary points (happens when the function is RSC), then there is a *HT-stable* stationary point $\tilde{\mathbf{x}}^*$ such that $f(\mathbf{x}^*) > f(\tilde{\mathbf{x}}^*)$ for any *HT-unstable* stationary point $\mathbf{x}^*$.

The following result provides sufficient conditions for the convergence of an IHT sequence. Corollary 5 aims to remove any restrictions on the initial condition. This corollary shows that no matter what initial condition in $C_s$ is selected, the IHT sequence will converge to a HT-stable stationary point. Note that we say a *HT-stable/unstable* stationary point $\mathbf{x}^*$ associated with $\gamma \in (0, \frac{1}{L_s}]$ is isolated if there exists a neighborhood $\mathcal{N}$ of $\mathbf{x}^*$ such that $\mathcal{N}$ does not contain any HT stationary point other than $\mathbf{x}^*$.

**Corollary 5.** *Let $f : \mathbb{R}^n \to \mathbb{R}$ be $L_s$-RSS and $C^1$ function. Suppose that every (nonempty) sub-level set of $f$ contained in $C_s$ is bounded. Consider an arbitrary $\gamma \in (0, \frac{1}{L_s}]$. Assume that*

    *A.1 : For any two distinct HT-stable stationary points $\mathbf{x}^*$ and $\mathbf{y}^*$ associated with $\gamma$, $f(\mathbf{x}^*) \neq f(\mathbf{y}^*)$. Then, for any $\mathbf{x}^0 \in C_s$, the IHT sequence $(\mathbf{x}^k)_{k \geq 0}$ converges to a HT-stable stationary point associated with $\gamma$. This convergence results also hold under the following assumption:*

        *A.2 : when $0 < \gamma < \frac{1}{L_s}$, each HT-stable stationary point associated with $\gamma$ is isolated.*

The following corollary shows that any IHT sequence always "escape" from a *HT-unstable* stationary point.

**Corollary 6.** *Let $f : \mathbb{R}^n \to \mathbb{R}$ be $L_s$-RSS and $C^1$ function. Suppose $f$ is bounded below on $C_s$. For any given $\gamma \in (0, \frac{1}{L_s}]$ and any HT-unstable stationary point $\mathbf{x}^*$ associated with $\gamma$, there exists a neighborhood $\mathcal{N}$ of $\mathbf{x}^*$ such that for any IHT sequence starting from any $\mathbf{x}^0 \in C_s$, there exists $N \in \mathbb{N}$ such that $\mathbf{x}^k \notin \mathcal{N} \cap C_s$ for all $k \geq N$.*

The next result shows the attraction towards a strictly HT-stable stationary point in a neighborhood of such a stationary point. In what follows, for each index subset $\mathcal{J}$ with $|\mathcal{J}| = s$, a subspace $\mathcal{S}_{\mathcal{J}} := \{\mathbf{x} \in \mathbb{R}^n \mid \mathbf{x}_{\mathcal{J}^c}\}$ associated with $\mathcal{J}$ is defined. Clearly, $C_s$ is the union of $\mathcal{S}_{\mathcal{J}}$'s for all $\mathcal{J}$'s with $|\mathcal{J}| = s$.

**Proposition 1.** *Let $f : \mathbb{R}^n \to \mathbb{R}$ be $L_s$-RSS and $C^1$ function. Suppose $f$ is bounded below on $C_s$ and $f$ is strictly convex on $\mathcal{S}_{\mathcal{J}}$ for any index subset $\mathcal{J}$ with $|\mathcal{J}| = s$. Let $\mathbf{x}^*$ be a strictly HT-stable*

*stationary point associated with any given* $\gamma \in (0, \frac{1}{L_s}]$. *Then there exists a neighborhood* $\mathcal{B}$ *of* $\mathbf{x}^*$ *such that for every* $\mathbf{x}^0 \in \mathcal{B} \cap C_s$, *the IHT sequence* $(\mathbf{x}^k)_{k \geq 0}$ *converges to* $\mathbf{x}^*$. *Moreover, if* $f$ *is strongly convex on* $\mathcal{J}$ *for every index subset* $\mathcal{J}$ *with* $|\mathcal{J}| = s$, *then for* $\mathbf{x}^0 \in \mathcal{B} \cap C_s$, *the IHT sequence* $(\mathbf{x}^k)_{k \geq 0}$ *Q-linearly converges to* $\mathbf{x}^*$.

Next, we provide an example to show the *escapability* property of *HT-unstable* points.

## 5 SIMULATION

To elaborate on theoretical results including Corollary 3, Theorem 2, the notion of *HT-stationary* points, Corollary 6 which shows *escapability* property of *HT-unstable stationary* points, and Proposition 1 which shows *Reachability* to *HT-stable stationary* points, we use a quadratic function $f(\mathbf{x}) = \frac{1}{m} \sum_{i=1}^{m} (A_{i\bullet}\mathbf{x} - y_i)^2 = \frac{1}{m} \|A\mathbf{x} - \mathbf{b}\|^2$ where $\mathbf{A} \in \mathbb{R}^{m \times n}$, $A_{i\bullet}$ is the $i$-th row of $A$, $\mathbf{x} \in \mathbb{R}^n$ is the optimization variable, and $\mathbf{y} \in \mathbb{R}^m$ is the target. This function is both RSS and RSC so both Corollary 6 and Proposition 1 follow. To better visualize the process, we let $m = n = 4$ and $s = 2$. Therefore, there are six *HT-stationary* points where the gradient over the nonzero elements is zero. We use Pytorch (Paszke et al., 2019) to select the matrix $A$ and $\mathbf{y}$. By setting the random seed to be 45966 we draw a $4 \times 4$ matrix $A$ whose elements are standard normal. Keeping the same seed, we generate $\mathbf{y}$. The following would be $A$ and $\mathbf{y}$:

$$A = \begin{bmatrix} -1.0655 & 0.2249 & -0.0897 & 0.1876 \\ 1.1627 & -1.1229 & -0.0823 & -0.3059 \\ -0.2011 & 0.5342 & -0.0551 & -1.3459 \\ 0.2308 & -0.6404 & -0.7468 & 0.0378 \end{bmatrix}, \quad \mathbf{y} = \begin{bmatrix} -1.7861 \\ -0.3556 \\ -0.1881 \\ 0.3896 \end{bmatrix}$$

The restricted Lipschitz constant, i.e., $L_s$, for the above quadratic function is $\frac{2}{m} \times \lambda_{max}(A^\top A)$ where $\lambda_{max}$ is the maximum eigenvalue of $A^\top A$. Thus, for the above choice of $A, \mathbf{y}$, the maximum allowable stepsize is $\gamma = \frac{1}{L_s} = 0.06$. Once, $\gamma$ is fixed, one can determine stability of each stationary point. The following are *HT-stationary* points along with their stability status as well as the gradient status of each *HT-stationary* point. As you can see, the gradient corresponding to nonzero elements in *HT-stationary* point are zero:

$$\begin{bmatrix} No. & x_1 & x_2 & x_3 & x_4 & g_1 & g_2 & g_3 & g_4 & HT-stability \\ 1 & 1.3474 & 1.0331 & 0 & 0 & 0 & 0 & 0.2060 & -0.3916 & \text{strictly HT-stable} \\ 2 & 0.6278 & 0 & 0.0177 & 0 & 0 & -0.3843 & 0 & -0.1070 & \text{HT-unstable} \\ 3 & 0.6387 & 0 & 0 & 0.1123 & 0 & -0.4189 & -0.0029 & 0 & \text{strictly HT-stable} \\ 4 & 0 & -0.1758 & 0.0008 & 0 & -0.6506 & 0 & 0 & 0.0106 & \text{HT-unstable} \\ 5 & 0 & -0.1776 & 0 & -0.0113 & -0.6473 & 0 & -0.0010 & 0 & \text{HT-unstable} \\ 6 & 0 & 0 & -0.1608 & 0.0259 & -0.7994 & 0.1297 & 0 & 0 & \text{HT-unstable} \end{bmatrix}$$

where $x_1, x_2, x_3, x_4$ are four coordinates of each *HT-stationary* point and where $g_1, g_2, g_3, g_4$ are the four gradient entries corresponding to each *HT-stationary* point. Since *HT-stationary* points are vectors in $\mathbb{R}^4$, there is no way to show all of them on one 2-d plane. Thus, we use six 2-d plains where each plane shows only two coordinates of *HT-stationary* points. On each 2-d plain we have 6 different points, each one associated with one of the *HT-stationary* points shown in a particular 2-d plain with specified coordinates. In Figure 1 the points with red stars are *HT-unstable* ones, and the blue ones are the *HT-stable* ones. For example, the first 2-d plain (first row-first column) including coordinates $x_1 - x_2$ shows the $x_1, x_2$ coordinates of all of the six *HT-stationary* points. On the first row-first column 2-d plain, the first *HT-stationary* point is more distinct because it is the only one that has two nonzeros elements associated with $x_1 - x_2$ coordinates. We also can see three points with $x_2 = 0$, two of which are *HT-unstable* points and one is *HT-unstable* one. This is more clear, if one looks at the column $x_2$ in *HT-stationary* points matrix above. Also, it is clear that we have three *HT-unstable* points with $x_1 = 0$ on the first 2-d plane.

We perturb nonzero coordinates of all *HT-unstable* points with a normal random noise with mean zero and standard deviation of $\sigma = 0.5$ to create 4,000 different initialization points. These points create four clouds around *HT-unstable* points which are shown in Figure 2. Then we run the IHT algorithm for 400 steps. After 300 steps, all of these initializations escape from those *HT-unstable* points and converge to either of the *HT-stable* stationary points on $x_1 - x_2$ or $x_1 - x_4$ 2-d planes. In fact, these



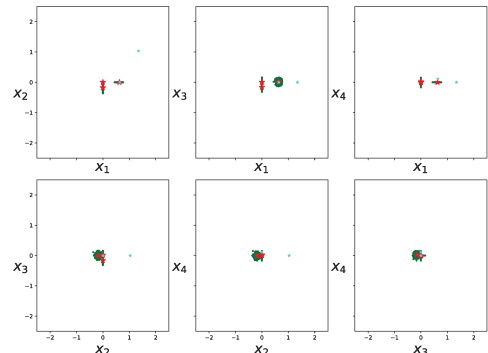

**Figure 1:** Illustration of *HT-stationary* points on 2-d plains.

**Figure 2:** Illustration of 4,000 initialization close to four *HT-unstable stationary* points.

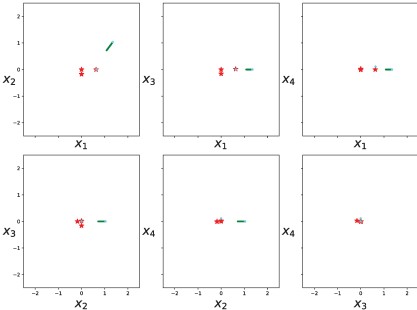

**Figure 3:** Illustration of 4,000 IHT sequences initialized close to four *HT-unstable stationary* points after 400 steps (please refer to the video of all 400 steps).

two *HT-stable* stationary points are sparse local minimizers. Figure 3 shows the 300-th step of IHT algorithm. There is a video in the supplementary materials that shows 400 steps of applying IHT algorithm for 4,000 different runs. These numerical results corroborate our theoretical results as expected. By looking at the video, one can easily see *escapability* property of *HT-unstable stationary* points, and *Reachability* to *HT-stable stationary* points.

## 6 CONCLUSION

This paper provide theoretical results that help to understand the IHT algorithm. These theoretical results include a critical gradient descent property of the hard thresholding (HT) operator which is used to show the sequence of the IHT algorithm is decreasing and by doing it over and over we get smaller objective value. This property also allows one to study the IHT algorithm when the stepsize is less than or equal to $1/L_s$, where $L_s > 0$ is the Lipschitz constant of the gradient of an objective function. We introduced different stationary points including HT-stable and HT-unstable stationary points and show no matter how close an initialization is to a HT-unstable stationary point, the IHT sequence leaves it. We provided a video of 4000 independent runs where the IHT algorithm is initialized very close to a HT-unstable stationary point and showed the sequences escape them. This property is used to prove that the IHT sequence converges to a HT-stable stationary point. Also, we established a condition for a HT-stable stationary that is a global minimizer with respect to $\gamma = 1/L_s$. Finally, we showed the IHT sequence always converges if the function values of HT-stable stationary points are distinct, this is a new assumption that has not been found in the literature.

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
