# OpenReview forum: "Gradient Properties of Hard Thresholding Operator"
_ICLR.cc/2023/Conference — Submitted to ICLR 2023_

### Official Review · Reviewer_BYNJ · 2022-10-24

**Confidence:** 5
**Correctness:** 4
**Technical Novelty And Significance:** 1
**Empirical Novelty And Significance:** 1
**Recommendation:** 1

**Clarity, Quality, Novelty And Reproducibility:**

The technical details are presented in a clear and clean way. The novelty of contribution, however, is not significant with respect to the prior best-known results. The overall quality of this work is below expectation.

**Strength And Weaknesses:**

# Strength:

 1. The main theoretical claims made in this paper are correct and mathematically sound.

2. The paper is in general well organized and the technical details are clearly presented .


# Weaknesses

1. The motivation behind this study looks not clear enough. The theoretical properties proved in the current work are fairly standard, and as admitted by the authors are well-addressed in the previous analysis of iterative hard-thresholding. I cannot see any particular reason why the present study about proving these prior results would be of any benefit for advancing the related lines of research.

2. The technical novelty of the main results is quite limited. More specifically, the gradient-norm descent property as summarized in Theorem 1 has already been developed in the proof of Jain et al. (2014, Theorem 1). With regard to the stability of hard-thresholding, some results identical to Theorem 2&3 have been established by Beck and Eldar (2013), and Yuan et al. (2014, Lemma 21)


# References:

P. Jain, A. Tewari, and P. Kar. On iterative hard thresholding methods for high-dimensional m-estimation. NIPS, 2014.

A. Beck and Y. C Eldar. Sparsity constrained nonlinear optimization: Optimality conditions and algorithms. SIOPT, 2013.

X. Yuan, P. Li and T. Zhang, Gradient Hard Thresholding Pursuit, JMLR, 2014.




**Summary Of The Paper:**

This paper presents a set of descent and optimality properties for the gradient hard-thresholding (GHT) operator widely used in sparse optimization algorithms such as iterative hard-thresholding. Regarding the descent property, the authors establish in Theorem 1 a gradient-norm descent property of GHT for restricted strongly smooth objectives. In terms of the optimality property, the authors mainly show in Theorem 2&3 that the HT-stability condition (as defined in Definition 4) is necessary for a global sparse minimizer, and for any accumulation point of an IHT sequence as well. A simulation study on sparsity recovery with quadratic loss is carried out to verify the theoretical predictions.

**Summary Of The Review:**

Based on the above comments, it is assessed that this paper in the current form is not strong enough to be published.

---

### Official Review · Reviewer_F4td · 2022-10-24

**Confidence:** 3
**Correctness:** 3
**Technical Novelty And Significance:** 2
**Empirical Novelty And Significance:** Not applicable
**Recommendation:** 3

**Clarity, Quality, Novelty And Reproducibility:**

This paper is generally well-written and easy to follow.

Some new technical results for the extensively-studied iterative hard thresholding (IHT) algorithm, although it is difficult to see how meaningful these new results are.

This work is mainly theoretical, and very limited simulation results are provided. Thus there is little concern about the reproducibility.

**Strength And Weaknesses:**

Strength:

This paper seems to be technically sound.

Weaknesses:

This seems to be a pure optimation paper. For example, almost all the closely relevant papers listed in Table 1 are published in optimization journals. In addition, the simulation results are weak and rather limited, and are not related to any practical machine learning/deep learning tasks.

The authors mention in the abstract that "the IHT sequence converges if the function values at HT-stable stationary points are distinct, where the last condition is a new assumption that has not been found in the literature". But from this paper, I cannot see how impactful/meaningful will this new assumption be.

Some minor comments:

In the abstract, using latex notation of the Lipschitz constant $L$.

Write down the full name of RSS when it first appears on page 3.

Claim 1 may be better written in ordinary text as it is a very simple and widely-used property, instead of highlighting it as a separate claim.

**Summary Of The Paper:**

In this paper, the authors study the extensively-studied iterative hard thresholding (IHT) algorithm. They establish a new and critical gradient descent property of the hard thresholding (HT) operator, which is related to the distance between sparse points. In addition, they introduce the notion of HT-stable/unstable stationary points, and establish the escapability property of HT-unstable stationary points and the local reachability property of strictly HT-stable stationary points. Moreover, they show that the IHT sequence converges globally under the assumption that the function values at HT-stable stationary points are distinct. Such an assumption is novel and has not been found in the literature. Finally, some simple simulation results are provided for synthetic data.

**Summary Of The Review:**

It is nice to see some new technical results for the iterative hard thresholding (IHT) algorithm. But it is a pity that from the current submission, I cannot see how meaningful/interesting these new results/assumptions are to the machine learning community.

From the current submission, my impression is that this is a pure optimization paper, and is better submitted to an optimization journal, rather than a machine learning conference.

---

### Official Review · Reviewer_kzNB · 2022-10-24

**Confidence:** 2
**Correctness:** 2
**Technical Novelty And Significance:** 2
**Empirical Novelty And Significance:** 2
**Recommendation:** 3

**Clarity, Quality, Novelty And Reproducibility:**

Clarity needs to be improved throughout the paper, and some statements should be more formal. Some examples are listed as follows:
1. In abstract, "the IHT sequence leaves it", "escape them".
2. In introduction, "the $\ell_0$-norm case have been"->"... has been".
3. In p.2 "$\mathcal{X}$ is a constraint set"
4. In (Q1), "for a local/global ..." -> "for the existence of a local/global ..."?
5. The description of Corollary 3 is entirely confusing and non-professional.
6. In p.8, "whose elements are standard normal"? Should that be about the probability distribution?

**Strength And Weaknesses:**

Strengths:
1. The paper has attempted to dig out theoretical properties of a classical hard-thresholding operator with some theoretical results.
2. The new concepts - HT-stable/HT-unstable stationary points - may be useful for other related discussions or studies.
3. The simulation results verified the proposed theories.

Weaknesses:
1. There are some minor language issues, and some statements are confusing.
2. It is not clear how the new concepts are valid and could be used for other studies.
3. The simulation section is not clearly organized. Since the data matrix is randomly generated, it would be great to check the average performance after many trials. No real applications are involved which would limit the applicability of theories.

**Summary Of The Paper:**

The paper studies the stability properties of the hard thresholding operator in gradient descent framework and introduces several new concepts such as HT-stable and HT-unstable stationary points. Moreover, the paper aims to answer four fundamental questions regarding the local/global minimizers and accumulation points of a sparse recovery problem. In the numerical experiments, the authors intend to use some synthetic data sets to justify the proposed theoretical results. However, the paper has limited novelty and lacks real data experiments from applications.

**Summary Of The Review:**

Although the paper proposes some new theories for discussing the gradient properties of hard-thresholding operators, the novelty and application outreach are limited.

---

### Official Review · Reviewer_raL8 · 2022-10-27

**Confidence:** 5
**Correctness:** 4
**Technical Novelty And Significance:** 2
**Empirical Novelty And Significance:** 2
**Recommendation:** 5

**Clarity, Quality, Novelty And Reproducibility:**

I will expand on novelty.

I believe that some of the results in Section 4.1 are similar to already known results. For example, a very similar property to what is proved in Theorem 1 and Corollary 2 can be seen in the inequality right before (15) in https://arxiv.org/pdf/2204.08274.pdf.

In addition, I believe the characterization of HT stationary points (also known as fixed points) in Section 4.2 is known. For example, in page 12 of https://people.maths.ox.ac.uk/thompson/SPARSpres.pdf the conditions are the same as Definition 4, so it is known that HT stationary points are the accumulation points of IHT.

The authors should explain in detail how these and similar previously known results relate to their work, since they seem to have a significant overlap.

**Strength And Weaknesses:**

Strengths:
- The theoretical analysis is sound.
- The results are presented in a clear and intuitive way.
- A numerical simulation of the IHT algorithm is presented.

Weaknesses:
- My main concern is novelty, which I will explain more about in the next section.
- The numerical simulation only consists of one random linear regression instance. It would be better to have an analysis multiple instances in order to account for the variance.

**Summary Of The Paper:**

This work studies the well known iterative hard thresholding (IHT) algorithm for sparse optimization. The algorithm works as follows: In each iteration of this algorithm, a gradient step is taken, and then the hard thresholding operator is applied, which restricts the solution vector to its top $s$ entries in absolute value. The results of the paper are convergence properties for smooth functions. Specifically, the authors prove that the algorithm converges to a stationary fixed point and characterize the properties of such fixed point.

**Summary Of The Review:**

In summary, the theoretical analysis is sound but I have concerns over its novelty that are not addressed by the manuscript. Therefore I cannot recommend acceptance until the contribution of this paper with respect to previous work is clearer.

---

### Decision · Program_Chairs · 2023-01-20

**Decision:**

Reject

**Justification For Why Not Higher Score:**

There were clear (and near-unanimous) concerns raised about novelty and significance, thus preventing the meta-reviewer from giving a higher score.

**Justification For Why Not Lower Score:**

N/A

**Metareview: Summary, Strengths And Weaknesses:**

The paper provides new results on the convergence properties of the iterative hard thresholding (IHT) algorithm for sparsity-constrained optimization, and its behavior around stable/unstable stationary points.

All reviewers (and the meta-reviewer) agree that the results are mathematically sound.

However, concerns were raised about the novelty and impact of the contributions. Many similar results (specifically, Theorem 1, Corollary 2, and Theorems 2/3) have appeared in several previous analyses of IHT. The assumption on isolated-ness of HT-stable stationary points made in Corollary 5 is hard to check, and in any case even if it were easy, the paper did not explain when this assumption is valid and/or useful. Moreover, the experimental section consisted of a single small linear regression scenario and it was not clear how the insights will generalize in practice. Moreover, the authors did not provide a response to any of the reviewers' comments.

Recommendation: reject.


**Summary Of Ac-Reviewer Meeting:**

N/A